# Innovative approach for potential scale-up to jump-start simplified management of sick young infants with possible serious bacterial infection when a referral is not feasible: Findings from implementation research

Abadi Leul[1]*, Tadele Hailu[1], Loko Abraham[1], Alemayehu Bayray[2], Wondwossen Terefe[2], Hagos Godefay[3], Mengesha Fantaye[4], Shamim Ahmad Qazi[5], Samira Aboubaker[5], Yasir Bin Nisar[6], Rajiv Bahl[6], Ephrem Tekle[7], Afework Mulugeta[2]

1 Department of Paediatrics and Child Health, School of Medicine, Mekelle University, Mekelle, Ethiopia, 2 School of Public Health, College of Health Sciences, Mekelle University, Mekelle, Ethiopia, 3 Tigray Health Bureau, Mekelle, Ethiopia, 4 Alamata Hospital, Alamata, Tigray, Ethiopia, 5 Department of Maternal, Newborn, Child and Adolescent Health, World Health Organization, Geneva, Switzerland, 6 Department of Maternal, Newborn, Child and Adolescent Health and Ageing, World Health Organization, Geneva, Switzerland, 7 Federal Ministry of Health, Maternal and Child Health Directorate, Addis Ababa, Ethiopia

* tsiabadi@yahoo.com

**Data Availability Statement:** All relevant data are within the manuscript.

## Abstract

### Background

Neonatal bacterial infections are a common cause of death, which can be managed well with inpatient treatment. Unfortunately, many families in low resource settings do not accept referral to a hospital. The World Health Organization (WHO) developed a guideline for management of young infants up to 2 months of age with possible serious bacterial infection (PSBI) when referral is not feasible. Government of Ethiopia with WHO evaluated the feasibility of implementing this guideline to increase coverage of treatment.

### Objective

The objective of this study was to implement a simplified antibiotic regimen (2 days gentamicin injection and 7 days oral amoxicillin) for management of sick young infants with PSBI in a programme setting when referral was not feasible to identify at least 80% of PSBI cases, achieve an overall adequate treatment coverage of at least 80% and document the challenges and opportunities for implementation at the community level in two districts in Tigray, Ethiopia.

### Methods

Using implementation research, we applied the PSBI guideline in a programme setting from January 2016 to August 2017 in Raya Alamata and Raya Azebo Woredas (districts) in Southern Tigray, Ethiopia with a population of 260884. Policy dialogue was held with decision-makers, programme implementers and stakeholders at federal, regional and district

**Funding:** This study was funded by the Bill and Melinda Gates Foundation through a grant to the World Health Organization. The funder had no role in study design, data collection and analysis, decision to publish, or preparation of the manuscript.

**Competing interests:** The authors have declared that there are no competing interests in this study.

levels, and a Technical Support Unit (TSU) was established. Health Extension Workers (HEWs) working at the health posts and supervisors working at the health centres were trained in WHO guideline to manage sick young infants when referral was not feasible. Communities were sensitized towards appropriate home care.

## Results

We identified 854 young infants with any sign of PSBI in the study population of 7857 live births. The expected live births during the study period were 9821. Assuming 10% of neonates will have any sign of PSBI within the first 2 months of life (n = 982), the coverage of appropriate treatment of PSBI cases in our study area was 87% (854/982). Of the 854 sick young infants, 333 (39%) were taken directly to a hospital and 521 (61%) were identified by HEW at health posts. Of the 521 young infants, 27 (5.2%) had signs of critical illness, 181 (34.7%) had signs of clinical severe infection, whereas 313 (60.1%) young infants 7–59 days of age had only fast breathing pneumonia. All young infants with critical illness accepted referral to a hospital, while 117/181 (64.6%) infants with clinical severe infection accepted referral. Families of 64 (35.3%) infants with clinical severe infection refused referral and were treated at the health post with injectable gentamicin for 2 days plus oral amoxicillin for 7 days. All 64 completed recommended gentamicin doses and 63/64 (98%) completed recommended amoxicillin doses. Of 313 young infants, 7–59 days with pneumonia who were treated by the HEWs without referral with oral amoxicillin for 7 days, 310 (99%) received all 14 doses. No deaths were reported among those treated on an outpatient basis at health posts. But 35/477 (7%) deaths occurred among young infants treated at hospital.

## Conclusions

When referral is not feasible, young infants with PSBI can be managed appropriately at health posts by HEWs in the existing health system in Ethiopia with high coverage, low treatment failure and a low case fatality rate. Moreover, fast breathing pneumonia in infants 7–59 days of age can be successfully treated at the health post without referral. Relatively higher mortality in sick young infants at the referral level health facilities warrants further investigation.

## Introduction

In 2018, an estimated 2.5 million neonatal deaths occurred worldwide, of which 1 million were in Sub Saharan Africa contributing to 36% and 47% of under-5 deaths respectively [1]. Neonatal infections primarily bacterial in origin, including pneumonia, sepsis and meningitis are one of the major causes of 55000 neonatal deaths every year [2]. An estimated 6.9 million episodes of possible serious bacterial infection (PSBI) in young infants up to 2 months of age occur yearly in low- and middle-income countries [3]. The rate of home deliveries is high; unhygienic conditions abound around birth and access to appropriate treatment for sepsis is low [4, 5]. WHO recommends inpatient treatment of neonates with infection/sepsis with injectable benzylpenicillin/ampicillin plus gentamicin as first-line antibiotics and supportive care [6]. Nevertheless, the referral acceptance rate remains low in many low- and middle-income country settings [7–13].

Observational studies from Bangladesh, India and Nepal have shown that neonatal infections could be managed by trained health providers when referral to a hospital is not feasible, subsequently reducing neonatal mortality [7–9]. Later, randomized controlled trials from Bangladesh, Democratic Republic of Congo (DRC), Kenya, Nigeria and Pakistan reported successful treatment of young infants up to 2 months of age with signs of PSBI with simplified antibiotic regimens at first-level health facilities when a referral was not feasible [10–13]. Using this evidence, the World Health Organization (WHO) developed a guideline to manage young infants with PSBI when referral is not feasible in 2015 [14]. It recommended both 2 and 7 days of injectable gentamicin plus 7 days of oral amoxicillin for treatment of young infants with signs of clinical severe infection (Box 1). In Ethiopia, due to difficult terrain and long distances to hospitals in rural areas, there was a need to provide access to treatment near people's homes. Government of Ethiopia instituted the health extension worker (HEW) programme in rural areas in 2003, where HEWs identify and provide care to sick children among other activities. The Government of Ethiopia trained HEWs to implement community-based newborn care which included PSBI management when the referral was not feasible in selected communities in three regions between 2008 to 2013, but the estimated treatment coverage was only around 50%. [15]. Some of the reasons for this low coverage were lack of awareness, perceived illness severity, perceived early treatment by mothers and having other young children [16]. Thus, the objective of this study was to implement a simplified antibiotic regimen (2 days gentamicin injection and 7 days oral amoxicillin) for management of sick young infants with PSBI in a programme setting when referral was not feasible to identify at least 80% of PSBI cases, achieve an overall adequate treatment coverage of at least 80% and document the challenges and opportunities for implementation at the community level in two districts in Tigray, Ethiopia.

### Box 1. Definitions and management strategies of young infants with signs of PSBI–adapted by Government of Ethiopia from WHO Guideline [14].

#### I. Signs of PSBI in a young infant 0–59 days of age

Severe chest indrawing, or no movement at all or movement only when stimulated, or not able to feed at all or not feeding well/stopped feeding well, or convulsions, or axillary high body temperature ($\geq 38°C$) or axillary low body temperature ($<35.5°C$) or young infant 0–6 days old with only fast breathing (respiratory rate $\geq 60$ breaths per minute),.

**Recommendation:** Refer immediately to a higher-level referral health facility

*If referral is not feasible, re-classify the sick young infant with signs of PSBI into the following*

**i) Critical illness:** A young infant who has any of the following signs—convulsions (fits) or is unable to feed at all or no movement at all

**Recommendation:** Refer immediately to a higher-level referral health facility. However, if referral is not feasible at all then treat the infant with injectable gentamicin once daily plus injection ampicillin twice daily up to 7 days on outpatient basis, with efforts made to refer the infant to a hospital as soon as possible.

**ii) Clinical Severe Infection (CSI):** A young infant who has any of the following signs— not feeding well/stopped feeding well, severe chest indrawing, high (38°C or above) or

low (less than 35.5˚C) axillary body temperature, movement only when stimulated, or fast breathing (60 or more breaths per minute) in infants 0–6 days of age (*this last component was added to CSI as local adaptation by Government of Ethiopia, instead of having it separately as severe pneumonia as designated in WHO guideline*)

**Recommendation:** Refer immediately to a higher-level referral health facility. However, if referral is not feasible then treat the infant with injectable gentamicin once daily for 2 days plus oral amoxicillin twice daily up to 7 days on outpatient basis.

## II. Sign of fast breathing pneumonia

Young infant 7–59 days old presenting with fast breathing (respiratory rate $\geq 60$ breaths per minute)

**Recommendation:** Treat with oral amoxicillin for 7 days without referral to a higher-level facility.

## Methods

### Context and setting of the study

This implementation research was conducted at the health post level in two districts, Raya Alamata and Raya Azebo in Tigray, one of the regional states of Ethiopia. The two districts are characterized by lowland and highland agro-ecological conditions. The total population of the two study districts was 260,884 [17]. The neonatal mortality rate in the state was 34 deaths per 1000 births in 2016 [18]. The two districts were selected based on their higher neonatal mortality rates compared to other districts in Tigray [19]. Raya Alamata has one general hospital, five health centres and 17 health posts. Raya Azebo has one primary hospital, seven health centres and 22 health posts. Each of the health posts is staffed with two female HEWs. These public health institutions provide maternal and young infant health services either free or at a low cost. The health system structure in Ethiopia is a three-tiered health system structure of primary care (primary health care unit), secondary care (general hospital) and tertiary care (referral hospital). The Primary Health Care Unit (PHCU) at the bottom level consists of health posts that serve 5000, health centers that serve 25000 and a primary hospital which serves 100000 people. The PHCU feeds into the secondary level of health care, i.e. general hospital, which serves a population of one million and the general hospital, in turn, feeds into the referral hospital which serves about 5 million people. The HEWs at health posts from the PHCUs carry out the 17 different activities in close collaboration with local Women's Development Groups (WDG) [20]. Among the main responsibilities entrusted to the WDGs is social mobilization for maternal, newborn and child health (MNCH) services. The HEWs are expected to conduct postnatal home visits on days 1, 3, 7 and 42 as per the MNCH guideline. A health centre has around 20 health professionals (health officers, nurses, midwives and laboratory technicians) but is not equipped to manage sick young infants on an inpatient basis.

### Study design

This implementation research was a longitudinal study, in which data were collected prospectively from each young infant aged up to 2 months in the catchment area from January 2016 to August 2017.

**Sample size determination.** The sample size was determined based on the proportion of new live births in a year (3.1%) in Ethiopia. Thus, it is expected that there will be about 8087 (3.1% x 260884) live births in a year in both districts. We assumed 10% of the young infants will develop PSBI and hence about 809 young infants were to be included in the implementation research in a year [11].

## Study population

All babies born during the study period were included in the study. Young infants up to 2 months of age with one or more signs of PSBI (Box 1) who were permanent residents of Raya Alamata and Raya Azebo districts were included in the study.

## Primary outcome

The primary outcome of this study was the identification of at least 80% of the PSBI cases and provision of appropriate treatment for at least 80% of those identified cases in young infants aged up to 2 months. Appropriate treatment was defined as the provision of inpatient treatment (injectable gentamicin and ampicillin or any other appropriate antibiotics) or outpatient treatment (oral amoxicillin twice daily for at least 5 days plus IM gentamicin injection for 2 days) to sick young infants with PSBI when the referral was not feasible.

## Secondary outcomes

Secondary outcomes included referral acceptance rate; infant death rate within two weeks after the initiation of inpatient or outpatient treatment; clinical deterioration defined as the emergence of any sign of critical illness or new sign of clinical severe infection during the outpatient treatment of sick young infants; any serious adverse effects such as local swelling at the injection site, new onset of rash, disseminated and severe rash, anaphylactic reaction, stopped passing urine for >12 hours (renal failure) and cellulitis or abscess at injection site in young infants who received simplified antibiotic treatment at health post level, rate of adherence to outpatient treatment and accuracy of classification of PSBI by HEWs.

## Planning the intervention

### Policy dialogue and consultative process

A policy dialogue facilitated by WHO was held between the Federal Ministry of Health (FMoH), the Regional Health Bureau (RHB), policy-makers, programme managers, health professionals, researchers from academia and other stakeholders. The WHO guideline for the management of PSBI when referral is not feasible [14] and evidence supporting it was presented and discussed [10–13]. The consensus was reached on interventions and on the need to test the implementation of the WHO PSBI guideline in a programme setting in Mekelle. Subsequently, consultations were held with stakeholders, including FMoH, Tigray RHB, WHO, UNICEF, Save the Children (an implementation partner of the RHB) and technical experts from Mekelle University to set up implementation research sites.

### Memorandum of understanding between TSU and stakeholders

Discussions were held among the TSU and key stakeholders to develop a memorandum of understanding, which articulated the roles and responsibilities of the RHB, Save the Children and the TSU (Table 1). The district health staff from the two selected districts were invited to participate.

**Table 1. Roles and responsibilities of the various stakeholders.**

| RN | Names of stakeholders | Roles and responsibilities |
|---|---|---|
| 1 | Technical support unit (TSU) from Mekelle University | • Leading the research activities |
| | | • Revision of the community-based neonatal care (CBNC) training materials to reflect the new WHO guideline [14] |
| | | • Supporting Save the Children, the implementing partner in training of health workers |
| | | • Orientation of women development group (WDG) members |
| | | • Monitoring and evaluation to assure quality |
| | | • Provision of technical support to Health Extension Workers (HEW), RHB and Save the Children |
| | | • Data management and analysis |
| | | • Dissemination of the progress of the implementation research through meetings and reports |
| | | • Document lessons and experiences regarding implementation of community-based treatment of possible serious bacterial infection (PSBI) |
| | | • Identify barriers and provide solutions to overcoming barriers. |
| 2 | Regional Health Bureau (RHB) | • Implementation of the PSBI management guideline in the selected districts |
| | | • Leading the quarterly review meetings at region/district levels |
| | | • Provide advices for adjustments as necessary and monitored progress through its regular reporting systems |
| | | • Fulfil the logistic and human resource needs |
| | | • Ensure health posts are open during working hours |
| | | • Mobilize HEWs and other health workers, as well as relevant stakeholders, to participate in trainings and review meetings |
| | | • Mobilize stakeholders to support the smooth implementation of community-based management of PSBI |
| | | • Conduct periodic supportive supervision |
| | | • Ensure the availability of essential CBNC supplies at the health posts and health centres. |
| 3 | Save the Children | • Implement the CBNC interventions, which included the training and post training follow-up of HEWs and other health workers |
| | | • Provide supportive supervision, performance review and clinical mentoring as per the CBNC protocol |
| | | • Ensure implementation issues identified during monitoring are communicated and acted upon in a timely fashion |
| | | • Support the implementation of strategies to improve newborn health care-seeking practices |
| | | • Report on the CBNC activities as per the nationally-agreed monitoring tools and indicators |
| | | • Share relevant information about training, and periodic monitoring and supervision |
| | | • Conduct joint supervision with the TSU and the RHB |
| | | • Participate in meetings at district/primary health care unit (PHCU) levels. |
| 4 | District Health Offices | • Ascertain the readiness of the implementation research sites |
| | | • Commit their time and human resources to support the implementation of the study in their respective communities |
| | | • Conduct joint supervision with the TSU, Save the Children and RHB |
| | | • Ensure the active participation of WDGs, maternal and child health experts, cluster supervisors and HEWs from the two districts |
| | | • Ensure the HEWs assessed, treated, referred and followed-up sick young infants and conducted scheduled postnatal visits |
| | | • Ensure the supervisors and clinical mentors visited the young infants being treated and provided technical support and on-site mentoring in addition to follow-up on days 5 and 8. |

## Sensitization meetings

A sensitization meeting in each district, facilitated by the TSU, was carried out with the study site administrators, women's affairs offices, WDGs, religious leaders, district health officers, hospital medical directors, health centre directors, supervisors and HEWs. Evidence for management of PSBI when referral is not feasible and its impact on neonatal mortality were discussed. This briefing increased trust with various stakeholders; led to buy-in and ownership of the study; improved linkages between various levels of the health system and resulted in smooth implementation of the research in the two study sites.

## Training of health care providers

To harmonize the conduct of the implementation research across the various sites, WHO organized a Master Training of Trainers workshop in Ibadan, Nigeria, for all implementation research sites including two sites in Ethiopia, two in Nigeria, one in DRC, one in Pakistan and four in India.

Two paediatricians from the TSU participated, and they subsequently trained master trainers from Save the Children and the RHB at the Tigray site. Chart booklets and training manuals on the Integrated Management of Newborn and Childhood Illness (IMNCI) were adapted from WHO [21] and translated to the local language (Tigrigna). Laminated job aids were also developed. The training was provided to the HEWs in two rounds in each district (woreda) and 15 to 20 trainees at a time attended. Refresher training took place every six months. The health centre directors, district health office experts, cluster supervisors and leaders of the 1 to 5 networks of mothers (a group of six neighbouring women led by a model household woman) were also trained.

## Interventions

HEWs from the health posts and nurses from the under-five clinics of the health centres were trained to provide health education, counsel mothers about newborn care including identification of signs of illness, and identifying and treating infants with PSBI (Box 1). HEWs identified sick young infants, assessed and classified them for PSBI, counselled families and provided treatment including to those who required referral but did not accept referral advice.

**Identification and management of patients.**   Amongst the roles and responsibilities of the health centre is to technically support the activities of the HEWs from the satellite health posts. The HEWs worked with the community-based WDG leaders to identify sick young infants and promote prompt care-seeking at the health posts. The sick young infants were identified either by the WDGs or HEWs during their house-to-house visits. When the WDGs identified sick young infants in their neighborhoods during the visits, they informed the HEWs either in person or via telephone. However, if a sick young infant was identified by the HEWs, the mother/caregiver was advised to take the sick young infant to the nearby health centre or hospital. If the mother/caregiver refused, the HEW provided treatment using the PSBI treatment protocol (Box 1). Sick young infants were also brought by the mothers/caregivers directly to the health post or the hospitals. Once at the health post, the HEW assessed and classified the sick young infant. If the sick young infant was found to have a clinical severe infection or critical illness, the HEW counseled the mother/caregiver on referral. If the mother/caregiver accepted the referral, the HEW gave a pre-referral antibiotic dose. If the mother/caregiver refused referral or the young infant was classified as having only fast breathing pneumonia, the infant was treated at the health post. Follow-ups were carried out by the HEWs and the nurse supervisors. To ensure that services were available at all times, one HEW made home visits while the other provided services at the health post. When an injection dose fell on a weekend or the family did not come for injection or follow-up, the HEWs made home visits.

**Referral mechanism.** According to FMoH policy, HEWs refer sick young infants with PSBI signs to their linked health centres. However, often health centres lack the capacity, commodities and infrastructure to admit and manage infants with PSBI. As a result, sick young infants are referred from the health centre to a hospital. During implementation, training was organized in consultation with the RHB to train health centre staff and HEWs to manage young infants with signs of PSBI with the simplified antibiotic regimens, when referral to a hospital was not accepted by the family.

**Quality control and assurance.** The clinical mentor provided on-site support to HEWs and supervisors every month. Additionally, on-call support for specific PSBI cases was given as required. TSU paediatricians and Save the Children facilitators also provided technical and clinical support when needed. The knowledge and skills of the HEWs in the identification, assessment, classification and treatment of young infants with PSBI were routinely monitored and evaluated by the paediatricians and clinical mentors every month, and on-site training to HEWs and supervisors was provided as needed. The research team used a standard checklist during its monthly field visits to monitor progress. Review meetings were conducted every three months with the TSU team, Save the Children, RHB, District Health Offices heads and experts, health centre heads, HEWs and cluster supervisors to discuss issues of concern for smooth implementation of the PSBI research. WHO and the FMoH also visited sites to review progress and monitor the quality of implementation. The classification and treatment selection of PSBI cases by the HEWs were reviewed and counter-checked for accuracy by the TSU paediatricians every month.

**Incentives provided to HEWs.** The HEWs from the health posts were given around US$ 2 for completing each case reporting forms (CRF), and the clinical mentor was given US$ 200 monthly to compensate for his/her time.

## Data collection and management

Data on pregnancy, birth, illness management, follow-up and outcomes were collected using pre-tested CRFs. HEWs collected and managed individual infant data with support from field supervisor nurses who checked the data regularly. Data from the field were sent to the study data management centre at Mekelle University. Inconsistencies were checked and resolved before data were entered into a computer using a specific database for the study. All young infants who were provided treatment were evaluated daily by the same treating HEW and by nurse supervisor on day 5 and day 8 for the ascertainment of the treatment outcome (improvement or deterioration).

## Data analysis

Quantitative data were cleaned and entered into SPSS version 20 software. After checking for missing data, outliers and invalid values, descriptive analysis was conducted to report frequencies, coverage and proportion of sick young infants with clinical severe infection, critical illness, pneumonia, local bacterial infections; treatment failures (clinical deterioration at any time and/or failure to respond on day 4 of treatment); referrals and treatment outcomes along with relevant information regarding PSBI management and treatment at the community level.

## Ethical considerations

The study protocol and all associated data collection instruments and consent forms were approved by the Ethical Review Boards of the College of Health Sciences at Mekelle University and Ethics Review Committee of WHO, Geneva. The study involved repeated visits to households. So, series of oral informed consents were obtained for participation in the study, i.e., for home visits during pregnancy and childbirth; for enrolment when the infant gets sick; for treatment; and for follow up visits.

## Results

We identified 7857 live births during the 20 months (January 2016 –August 2017) of the study. Home visits by HEWs were made to 1790 (23%) on day 1, 5140 (65%) on day 3, 6376 (81%) on

day 7, 6526 (83%) on day 42 and 6713 (85%) on day 59. We identified 854 infants with any sign of PSBI during the first two months of life.

## Coverage and management of PSBI

We estimated that 25% of live births were not identified by HEWs in this study, particularly in the early part of the implementation (n = 1964), similar to that reported in a similar study from Zaria, Nigeria [22], leading to estimated total live births of 9821 during the study period. Assuming 10% of infants will have any sign of PSBI within the first two months of life (n = 982) [11], the identification and coverage of treatment of PSBI cases in our study area was 87% (854/982).

Of the 854 cases, 333 (39%) young infants were directly brought to the hospital by families and were treated there, whereas 521 (61%) were seen by HEWs or nurses. The coverage of identification of PSBI by the HEWs improved from 37% in the first quarter to 78% in the last quarter. (Fig 1). Of the 521 PSBI cases identified by HEWs and nurses, 313 (60.1%) young infants 7–59 days old had only fast breathing pneumonia, and all of them were treated with oral amoxicillin either at a health post or health centre without a referral (Table 2). One child did not respond to oral amoxicillin, and two were lost to follow-up. No serious adverse effect or death was reported.

Of these 521 infants with any sign of PSBI, 181 (34.7%) had signs of clinical severe infection, while 27 (5.2%) had signs of critical illness. Of the 181 young infants with clinical severe infection, the caregivers of 117 (64.6%) accepted referral to a hospital for treatment, whereas the caregivers of 64 (35.4%) refused referral and were treated at a health post. All except one (63, 98%) who were treated at health post received appropriate treatment. One was lost to follow-up, but no death was reported. Treatment adherence was 100% for injection gentamicin and 98% for oral amoxicillin respectively (Table 3). All cases of critical illness (n = 27) identified by HEWs accepted the referral and were treated in a hospital (Fig 2).

## Accuracy of classification of HEWs

TSU paediatricians checked 390 data forms (182 with fast breathing pneumonia, 181 with clinical severe infection and 27 with critical illnesses) filled out by the HEWs and compared their

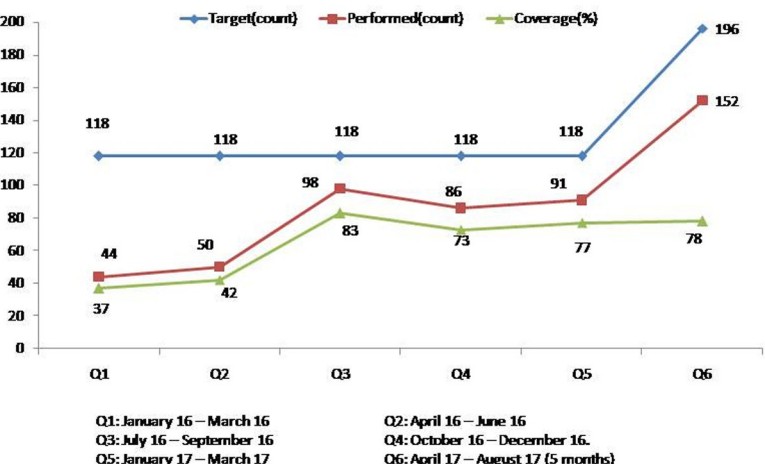

**Fig 1. Identification of PSBI cases against target by HEWs and nurse supervisors in each quarter of study, January 2016 to August 2017, n = 521.**

**Table 2. Young infants 7–59 days of age with only fast breathing pneumonia identified at health posts and health centres (n = 313).**

| Parameters | N (%) |
|---|---|
| Identified and treated on outpatient basis at either health posts or health centres | 313 (100) |
| Identified at health posts by health extension workers | 182 (58) |
| Identified at health centres by nurses | 131 (42) |
| Completed treatment | 310 (99) |
| **Compliance to treatment** | |
| Received all 14 doses of amoxicillin | 308 (98) |
| Received 10–13 doses of amoxicillin | 2 (0.6) |
| Received 6–9 doses of amoxicillin | 1 (0.3) |
| Missing data | 2 (0.6) |
| **Follow-up of infants** | |
| Completed all follow-up visits | 308 (98) |
| Partially followed-up (all follow-up visits not completed) | 2 (0.6) |
| Lost to follow-up (outcome unknown) | 2 (0.6) |
| **Treatment outcomes**[*] | |
| Declared as 'clinical treatment success' | 310 (99) |
| Declared as 'clinical treatment failure'–developed signs of severe illness | 1 (0.3) |
| Infants 7–59 days with fast breathing only with outcome unknown | 2 (0.6) |

[*]No death was documented.

recorded signs with their classifications and treatment. Only 22 of the 182 (12%) young infants with fast breathing and 11 of the 181 (6%) with signs of clinical severe infection were misclassified. None of the young infants with signs of critical illness was misclassified. Classifying young infants < 7 days of age with only fast breathing pneumonia was the most common misclassification by the HEWs. However, no discrepancy was detected in treatment selection for the classification of the young infant.

## Deaths

Sick young infants with PSBI were followed-up on daily basis by the treating HEW and on day 5, day 8 and day 14 by the nurse supervisor when treatment was provided as an outpatient in the HP. Those young infants who were treated at a hospital were visited on day 14 to record their outcomes. Out of 854 young infants with PSBI, 35 (4%) died. All deaths occurred among infants who were treated at a hospital, 18 at Mehoni and 17 at Raya Alamata. Seven of these deaths were amongst the 144 (117 with clinical severe infection and 27 with critical illnesses) who were referred by HEWs (three had a critical illness and four had a clinical severe infection) to a hospital. All were given pre-referral treatment.

## Implementation challenges and solutions

During this implementation research, several challenges were encountered. Solutions were found for most of them through collaborative efforts between TSU, the RHB, Save the Children, HEWs, health centres, and the district health office (Table 4).

## Discussion

Our data show that the management of PSBI when the referral is not feasible is possible in the Ethiopian context. Coverage of treatment was higher (87%) than our target, which showed

Table 3. Young infants 0–59 days of age with signs of clinical severe infection (n = 181).

| Parameters | N (%) |
|---|---|
| Brought by families to health post and classified as clinical severe infection | 181 |
| Referred to hospital | 181 (100%) |
| Accepted referral to hospital | 117 (64.6) |
| Did not accept referral | 64 (35.4) |
| Accepted treatment at health post | 64 (100) |
| Completed treatment | 63 (98) |
| **Compliance to treatment** | |
| Received 2 injections of gentamicin | 64 (100) |
| Received all 14 doses of DT amoxicillin | 62 (96.8) |
| Received 10–13 doses of DT amoxicillin | 1 (1.6) |
| Missing data | 1 (1.6) |
| **Follow-up of infants** | |
| Completed all follow-up visits | 63 (98) |
| Lost to follow-up (outcome unknown) | 1 (1.6) |
| **Treatment outcomes***  | |
| Declared as 'clinical treatment success' | 63 (98) |
| Declared as 'outcome unknown' | 1 (1.6) |

*No death was documented.

good utilization of services. In those who were treated on an outpatient basis, appropriate treatment and treatment completion rates (98%) were also very high. The fast breathing pneumonia in young infants age 7–59 days, which comprised of about two-fifth of all PSBI cases were treated successfully on an outpatient level without a referral. In those treated on an outpatient basis, a low treatment failure rate and no deaths were documented. These findings are in agreement with the findings from AFRINEST study conducted in similar communities from Nigeria, DRC and Kenya [11].

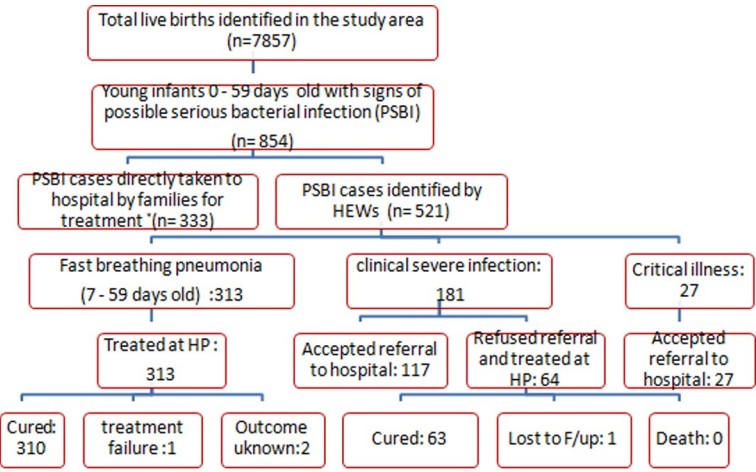

**NB**: 35/477 (7.3%) died among those treated at a Hospital

**Fig 2. Schematic representation of the identification, assessment, classification, referral, treatment and follow up of sick young infants.**

**Table 4.  Implementation challenges encountered and solutions provided.**

| Stages | Implementation challenge | Solutions and actions |
|---|---|---|
| Patient identification, referral, treatment and follow up | Poor pregnancy surveillance, recording and poor linkage between antenatal care and the health post and health centre/hospital | A format on pregnancy cohort (listing pregnant mothers whose expected date of delivery was in the same month separately) was designed and distributed to each of the health centres. On-site training was provided to HEWs on the use of the format by the TSU. Health centres linked the pregnant women who visited the health centre for their first antenatal care follow-up with the HEWs. Laboratory results of pregnant women were communicated through a formal referral slip to the HEWs from the health posts. |
| | Poor postnatal care home visit services | Consultations and discussions were held between the HEWs and the TSU to ensure adherence to the postnatal care guideline of the FMoH (home visits on days 1, 3, 7 and 42). Sustainability remains a challenge. |
| | Failure of HEWs to carry out CBNC activities properly | Refresher training was provided to HEWs by master trainers and Save the Children with support from the TSU. This was complemented with on-site training by the TSU, the clinical mentors and the supervisors during the first two quarters which resulted in improvement in the skills to carry out activities as per standard case management. |
| | Mismatch between assessment and classification as per chart booklet (not using the chart booklet) | On-site training was provided to HEWs by the TSU on the use of the job aids distributed to health posts during the first quarter, with subsequent on-site mentorship by the TSU through the second and third quarters. This resulted in improvement in the skills to carry out activities as per standard case management in the subsequent quarters. |
| | Poor on-site supportive supervision from the district health office and cluster supervisors | Consultations, review meetings and discussions were held with the RHB, district health office and health centre staff and HEWs. Training was provided to the district supervisors and health centre staff by the TSU. Subsequently agreements were reached between the TSU, health centres and the District Health Offices to increase the frequency of supportive supervisory visits to health posts from once in six months to at least quarterly by the district health office and monthly by the TSU. |
| | Confusion in the definition of young infant as a neonate (first 28 days of life) | Refresher and onsite training were provided to HEWs, health centre and district health staff by the TSU to resolve confusion. |
| | Confusion between integrated community case management (iCCM) and CBNC during the early days of project implementation | HEWs, health workers from health centres and experts from the districts had a better awareness of iCCM than CBNC as iCCM had been in place for a few years. Refresher and on-site training were provided by Save the Children and the TSU. |
| | Poor referral linkage across all levels | Consultations and discussions were held between the TSU, FMoH, RHB and the district health office to improve the referral and feedback linkages and communication between health workers at different levels. They agreed to have an auditable referral system and to review the referrals during their meetings. |
| | Discrepancies between the management of PSBI at health posts and health centres and ill-equipped health centres to manage referred PSBI cases | Refresher and on-site training during the supportive supervisory visits were provided to health post and health centre staff in the study area. In addition, consultations and discussions were made with the RHB to address discrepancies. As a result, the RHB wrote a letter to the health centres to comply with using the simplified antibiotic treatment regimen implemented by the health posts unless the health centre has the capacity to admit and manage cases on an inpatient basis. The health centre workers were trained on PSBI case management by the TSU during the third quarter. |
| | Poor health care-seeking behaviour of mothers | Refresher training was provided to the HEWs and WDGs to create demand for the services provided at the health post. Community mobilization activities to increase care-seeking were also undertaken. |

(*Continued*)

**Table 4.** (Continued)

| Stages | Implementation challenge | Solutions and actions |
|---|---|---|
| Referral and treatment | Tendency of the HEWs to treat rather than to refer sick young infants with PSBI during the initial period of implementation research | Refresher training and on-site supportive supervisory visits were provided to the HEWs to counsel mothers or families of sick young infants for referral to the next higher level of care. HEWs were made to understand to only treat sick young infants whose families refused to accept referral. |
| | Expiry of medicines for simplified treatment regimen of PSBI | During the on-site supportive supervisory visits of the TSU, the HEWs from the health posts were trained on how to read the expiry dates of drugs, report the drugs with short shelf lives and redistribute them to nearby health posts. Templates were developed for redistribution and reporting of these drugs in consultation with district health offices. |
| Data Collection | The health management information system format was not capturing data on PSBI | TSU held discussions with the RHB and the FMoH to align the information in the health management information system and CBNC, which resulted in a few key indicators to be incorporated in the revised DHIS. |
| | Complicated data collection instruments (CRFs) has challenged their completion | During the initial period of project implementation, the HEWs were concerned about the number and complexity of the data collection tools and did not complete some of the forms. Additional onsite training was provided to the HEWs to complete the various CRFs. TSU also revised the tools to make them much simpler. |
| Administrative issues | Heavy workload of HEWs and missing of scheduled visits | Consultations and discussions were made with the RHB to increase the number of HEWs. The RHB agreed in principle, but the number did not increase in many places. Health centre staff provided support to health posts when HEWs were unavailable, which this has worked well. |
| | Disconnect between the health posts and the district health office and health centres regarding the implementation of the CBNC programme | Consultations and discussions were held between the TSU and the district health office and health centres on CBNC and health post-based management of PSBI during the end of the first quarter, and quarterly review meetings chaired by the head of the RHB were held to improve the buy-in and ownership of the CBNC by the district health office and health centres. The district health office supervisors were included in the monitoring and evaluation activities of the TSU starting in the second quarter. |
| | Closure of health posts during working hours, especially during campaigns | Discussions were held with the district health offices and the HEWs in order to have one HEW at the health post during working hours. Additionally, it was agreed that the health posts would remain open during working hours even during public health campaigns. The district health office committed to send health workers from health centres when both the HEWs were unavailable at the health post. |

The referral acceptance rate (65%) among young infants with clinical severe infection was high in our study. We believe the main reasons for the better utilization of services and higher referral acceptance was most probably the presence of relatively active WDGs and their leaders to some extent especially after the initial implementation phase, free ambulance service and free treatment of neonates at the receiving health facilities. It resulted in better utilization of CBNC and PSBI services and building trust and confidence in the HEWs and health posts to treat and manage newborn illnesses at the community level when the referral was not feasible. On the other hand, the barriers for referral acceptance were long travel distance, low health-care-seeking behaviour and cultural barriers as postpartum mothers do confine themselves in their homes till the time of baptism (40 days and 80 days for boys and girls respectively). Other studies have reported various reasons for refusal of referral advice such as economic constraints, distance to the hospital, quality of care or attitude of the health workers, poor referral system and lack of transport, cost of travel and treatment, lack of permission from family members, religious and cultural beliefs, and issues with lack of child care and other logistical problems [23–26].

When we compare our data with other PSBI implementation research studies in various countries, we find some similarities and some differences. Our treatment coverage of 87% was a bit lower than that reported by the study in Zaria, Nigeria (96%) [22]; but was higher than those reported from Malawi (64%) [26]; Lucknow, India (53%) [27] and Kushtia, Bangladesh (31%) [28]. Like our study, high treatment completion rates were also reported by Malawi (95%) [26], Zaria, Nigeria (94.1%) [22] and MaMoni project, Bangladesh (80%) [29] in infants with clinical severe infection treated on an outpatient basis. Our proportion of fast breathing pneumonia in 7–59 day-old infants was 60% among those who were identified at a health post or a health centre, compared to 87% in Kushtia, Bangladesh [28], 30% in MaMoni project, Bangladesh [24], 28% in Malawi [26], 22% in Zaria, Nigeria [22] and 13.3% in Lucknow, India [27].

To effectively implement PSBI case management, an important step is to track all pregnant facilitated postnatal home visits, especially in the latter part of the implementation. The strategy of WDGs was conceptualized as a way to identify pregnancies and births, create demand for health care, wellness, and improve access to health care [30]. The close working relationship between the HEWs and WDGs has proven to be an efficient and effective approach to improve women and births in settings similar to the study communities [31, 20]. The involvement of the WDGs in the identification of cases, those who were lost to follow-up, and pregnant women who need close attention and immediate referral can improve the links between WDGs and HEWs and eventually improve care-seeking behaviour of families.

Although postnatal home visits by a trained health worker is an effective intervention to reduce neonatal mortality [32–34], less attention was given to these in our study areas. The overall proportion of young infants visited within two days of their birth has been reported to be around 16% in Ethiopia [35]. As neonatal mortality is the highest in the first week of life, timely provision of postnatal care services and prompt identification and appropriate care-seeking for sick young infants helps in reducing neonatal mortality. Large numbers of women and their newborns remain at home during and immediately after birth, so building and reinforcing the links between the community and health facilities is essential to improve the provision of postnatal services. Strengthening this link will be key for the successful implementation of PSBI case management at a community level in the Ethiopian setting. Thus, as a complementary strategy to improving PNC home visits, reinforcing and strengthening the links between the health extension program and the community through the active participation of the WDGs (women development groups) and the links with the midwives who support the skilled deliveries of mothers in the health facilities (health centres/hospitals) is highly recommended.

During the first three months of the implementation period, there were discrepancies in the management of PSBI cases at the health centres and health posts. As health centres did not have facilities for admission, staffs were trained to treat sick young infants with PSBI when a referral was not feasible with twice ampicillin and once gentamicin injections for seven days on an outpatient basis. This was practically difficult, and thus almost all newborns with PSBI that used to come to the health centre ended up being referred to hospital thereby creating unnecessary delay. This issue was brought to the attention of the RHB, and it was agreed that the management of PSBI cases should follow the same protocol as that followed by HEWs. Following this decision, the health centre staffs were trained in PSBI management according to the new guideline. Data regarding PSBI activities were not captured on the District and Health Information System (DHIS) and resulted in discrepancies in what was being reported and done. After discussion with regional and federal health authorities, it was agreed to incorporate a few key indicators regarding PSBI case management in the DHIS. Besides, there were

discrepancies between DHIS and CBNC data collection formats, which needed an alignment and due attention and action by the health sector.

The TSU presented the implementation process and the findings together with its challenges and successes to the FMoH and RHB. The national child health technical working group deliberated upon the findings from Tigray and the other study site in Jimma and recommended i) outpatient management of fast breathing pneumonia in a young infant aged 7–59 days with oral amoxicillin without referral and ii) use of gentamicin injection for two days plus oral amoxicillin for 7 days for management of clinical severe infection when a referral is not feasible. These have been now accepted as policy by the government and national IMNCI materials have been revised and are being scaled up in the country [36].

Our findings are sustainable in the absence of the study support and incentives because the implementation was entirely done by the existing health system with modest refresher trainings. These refresher trainings can easily be conducted by the health work force from the nearby health facilities (health centres and hospitals). Moreover, they are generalizable to the Ethiopian setting because the study communities are similar and the CBNC programmes are implemented following the same strategies. Besides the study data tools were adapted from the CBNC register which is used by HPs throughout the country.

There were several keys to successful intervention implementation in our study. First, high acceptance of referral advice and self-referral of sick young infants by families to the hospitals themselves. Second, provision of technical support to HEWs and other workers by expert paediatricians in the TSU and the ability of the TSU to conduct a dialogue with policy-makers for policy revisions and updates. Third, a common understanding of policy-makers (RHB, FMoH) and researchers about the identification and management of PSBI and a common objective of identifying viable solutions to increase access and coverage. The political commitment, health policy of the country, organization of the community and organization of the health care services were enabling factors for successful implementation. Fourth, this implementation research created a platform for discussion on optimization of care at different levels of the health system. The ability of health centres to function as referral sites for HEWs still warrants serious discussion. Finally, the partnership between technical experts, policy-makers and implementers to achieve the same objective was very fruitful. The support from the TSU contributed to the identification of implementation bottlenecks and facilitated problem-solving.

Our study had a couple of limitations, including the discrepancy between the expected and identified number of live births. The absence of direct baseline data before the intervention was another limitation of this study. The links between WDGs and HEWs were not as strong as expected, resulting in poor care-seeking behaviour among the mothers of young infants particularly in the initial stages of the implementation that resulted in relatively low cases in the initial periods of the implementation. The high turnover of the HEWs is another limitation, which is the result of high workload, absence of career structure in the health sector for HEWs and lack of incentives. The situation was felt during the monthly field visits as we were meeting new faces of HEWs. To address the high turnover of the HEWS, increasing the number of HEWs per health post (at least 3), creating career structure for HEWs and the construction of residential houses within the premises of the health posts were discussed with the health authorities to limit the fast turnover. Nonetheless, findings are likely representative of patterns of PSBI case management in rural Ethiopia.

In conclusion, PSBI management at the community level was fully implemented in all the health posts from the two districts with technical assistance from the TSU. The provision of training on assessment, classification and treatment of young infants with PSBI, together with regular mentoring and supervision from the TSU and supervisors and availability of supplies/

commodities made HEWs confident to manage PSBI in young infants when a referral was not feasible. The treatment success rate was high with no deaths among PSBI cases managed by HEWs at health posts. Also, young infants 7–59 days with only fast breathing were successfully treated at the health post without a referral, thus increasing the access to treatment to a large proportion of sick young infants. Mortality occurred at the hospital level rather than at the health post level which might be due to sicker infants being referred to a hospital, delay in reaching the hospital or potentially hospital-acquired infections. Potential delays in a presentation at the hospital and quality of care at referral institutions need urgent attention by regional and federal health authorities. In Ethiopia, where neonatal mortality is still very high, management of PSBI when a referral is not feasible is essential. Our data show that it can be implemented at the primary health care level within the existing health system. Finally, the TSU contributed to solving challenges and finding solutions to implement the new guideline successfully. During scale-up, regular mentoring and supportive supervision of health services staff will be essential to achieve the desired goal.

## Acknowledgments

The authors are thankful to the study participants and data collectors, specifically the HEWs from Raya Alamata and Raya Azebo Districts. We acknowledge support from the RHB, District Health Offices, health facilities (hospitals, health centres and health posts), FMoH, Save the Children, WHO Headquarters and Mekelle University.

## Author Contributions

**Conceptualization:** Abadi Leul, Shamim Ahmad Qazi, Samira Aboubaker, Rajiv Bahl, Ephrem Tekle, Afework Mulugeta.

**Formal analysis:** Abadi Leul, Tadele Hailu, Alemayehu Bayray, Wondwossen Terefe, Shamim Ahmad Qazi, Samira Aboubaker, Yasir Bin Nisar, Afework Mulugeta.

**Investigation:** Abadi Leul.

**Methodology:** Abadi Leul, Alemayehu Bayray, Shamim Ahmad Qazi, Samira Aboubaker, Rajiv Bahl, Afework Mulugeta.

**Project administration:** Abadi Leul, Loko Abraham, Alemayehu Bayray, Mengesha Fantaye, Afework Mulugeta.

**Supervision:** Abadi Leul, Tadele Hailu, Alemayehu Bayray, Wondwossen Terefe, Hagos Godefay, Mengesha Fantaye, Shamim Ahmad Qazi, Samira Aboubaker, Ephrem Tekle, Afework Mulugeta.

**Validation:** Alemayehu Bayray.

**Writing – original draft:** Abadi Leul, Tadele Hailu, Alemayehu Bayray, Shamim Ahmad Qazi, Samira Aboubaker, Yasir Bin Nisar, Afework Mulugeta.

**Writing – review & editing:** Abadi Leul, Tadele Hailu, Alemayehu Bayray, Hagos Godefay, Shamim Ahmad Qazi, Samira Aboubaker, Yasir Bin Nisar, Rajiv Bahl, Afework Mulugeta.

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
