## [Decision Letter · Decision Letter 0]

26 Jun 2020

PONE-D-20-08171

Innovative approach for potential scale-up to jump-start simplified management of sick young infants with possible serious bacterial infection when referral is not feasible: findings from implementation research

PLOS ONE

Dear Dr. Luel,

Thank you for submitting your manuscript to PLOS ONE. After careful consideration, we feel that it has merit but does not fully meet PLOS ONE’s publication criteria as it currently stands. Therefore, we invite you to submit a revised version of the manuscript that addresses the points raised during the review process.

We look forward to receiving your revised manuscript.

Kind regards,

Lawrence Palinkas

Academic Editor

PLOS ONE

Journal Requirements:

Reviewers' comments:

Reviewer's Responses to Questions

**Comments to the Author**

1. Is the manuscript technically sound, and do the data support the conclusions?

Reviewer #1: Yes

Reviewer #2: Partly

Reviewer #3: Partly

2. Has the statistical analysis been performed appropriately and rigorously? 

Reviewer #1: N/A

Reviewer #2: No

Reviewer #3: No

3. Have the authors made all data underlying the findings in their manuscript fully available?

Reviewer #1: Yes

Reviewer #2: Yes

Reviewer #3: Yes

4. Is the manuscript presented in an intelligible fashion and written in standard English?

Reviewer #1: Yes

Reviewer #2: Yes

Reviewer #3: No

5. Review Comments to the Author

Reviewer #1: This study is an important contribution as it provides evidence of the feasibility of implementing the PSBI guideline in Ethiopia and identifies challenges and their solutions. However, the manuscript requires revision and restructuring to be ready for publication (particularly the methods section).

Specific suggestions and questions are provided by section below:

METHODS:

Context and setting:

• Correct discrepancy – last sentence of introduction states: “in two regions in Ethiopia, Tigray and Jimma” while methods mention only two districts of Tigray (nothing in Jimma). Issue occurs again in last sentence of data management section: “Common data collection tools and the database were used by both the Tigray and Jimma sites”

• Would be useful to provide readers a bit more information on characteristics of HEWs (female, education and training levels). In addition, 17 activities of HEWs mentioned but not described – suggest to give examples and/or provide reference for people to have more info on HEWs and their roles

• Health system is well-described but there is little information provided on the communities (beyond the population size) – recommend to also provide short description of the community socio-demographics

Study population: the study tracked all live births during the study period to determine the denominator for estimated coverage – therefore the study population should be expanded to indicate that all babies born during the study period were included in the study and differentiate between those infants that were identified as sick and then treated through the study.

Primary and secondary outcomes: these are not sufficiently described or defined. Each primary and secondary outcome should be defined, ideally in a table. In addition, secondary outcomes should be expanded to include other indicators that are reported on in the results (e.g. follow-up, accuracy of HEWs)

Sample size determination is missing – should be added to Methods section.

Intervention: this section repeats information on the health system covered on content and that could be removed. Information on follow-up schedule (when, where, by who) should be included here.

Data management: should be revised to “Data collection and data management” and include information on the schedule for data collection (Day 3, Day 7? Etc)

Implementation phases:

- Phases of implementation: This section is very long and while important, could be condensed using tables with additional information provided in supplementary files. Specifically – recommend covering roles and responsibilities of each stakeholder group in a table (either in paper or supplementary file)

- Implementation Phase: it would help the reader to include a figure showing how cases of sick young infants were identified, referred, treated and followed up.

- Some content under patient identification should be included in results or discussion section – specifically: “In the initial stages, the demand side was weak which improved, particularly in those areas where WDGs and their leaders were more active. It resulted in better utilization of CBNC and PSBI services and building trust and confidence in the HEWs and health posts to treat and manage newborn illnesses at the community level when referral was not feasible.”

- Incentives– information on incentives to HEWs is included under the “Role of the District Health Office” but should be included under the “Implementation Phase” with its own sub-heading. Any other incentives provided through the study should be described here as well.

RESULTS:

- A table with results for the 117 children that accepted referral to the hospital and the 333 children who went directly to the hospital should be included (if not possible should be explained why not)

- Implementation challenges and solutions – a long list is provided, and it would be helpful for the reader to categorize or group these challenges somehow (perhaps into what stage – patient identification, referral, treatment, follow-up)

DISCUSSION:

- First sentence should include references to identify the studies referred to

- What are the reasons that acceptance of referral was so high in this setting? This contrasts with many other studies and deserves some attention in the discussion – particularly given the mortality that occurred there.

- The sentence” To effectively implement PSBI case management, the first step is to track all pregnant women and births” is misleading – it is possible to provide PSBI case management without this (as per Malawi example), but facility delivery levels and care-seeking must be high – so should be reworded to be specific to this context

- Study recommends strengthening postnatal home visits - what specifically should be done and how effective are these likely to be. The literature shows that obtaining high coverage of PNC home visits has not been successful in most settings (<20% coverage)– suggest referring to analysis by McPherson (https://www.ncbi.nlm.nih.gov/pmc/articles/PMC6005634/) and providing potential alternative/complementary strategies to increase identification of sick young infants

- High turnover of HEWs is mentioned as a limitation only in passing but would be quite important to understand – this should be elaborated on (what level of turnover, what possible reasons and what might be done to limit it)

- Discussion should address how generalizable are the findings to other settings in Ethiopia and how likely these results are to be sustained in the absence of the study support and incentives

Minor comment: Paper should receive a careful final edit as there are several typos and incomplete sentences.

Reviewer #2: Please see full attached comments.

General issues:

-terminology is interchanged and unclear throughout. What is the difference between sick young infants, PSBI, critical illness and clinical severe illness? These terms need to be defined and using distinctly.

-Unclear whether this study is looking at infants <2 months or 7-59 days as both are used

-More sophisticated data analysis should be used. For quality improvement/implementation research, a run chart or statistical process control chart (preferred) is necessary to demonstrate significant change due to interventions.

-There is no statistical significance reported for data analysis. The data is reported without any analysis provided. Overall rates are reported but there is no comparison to baseline data or trends over time. This is not standard for implementation research and prevents any conclusions to be made in terms of effectiveness of the interventions.

-There is no comparison of post-implementation data to a baseline data set. Baseline rates reported as “low” but the data are not reported. This prevents any conclusions that the interventions were successful.

-A key driver diagram is essential to a quality improvement/implementation project. This should be included to demonstrate the issues/barriers and linkage with interventions to achieve aims.

-The discussion of the interventions and practical strategies for implementation is strong. However, the data analysis is not strong enough to support conclusions of success. This manuscript is more descriptive of intervention implementation and barriers/solutions than true quality improvement research.

-There are so many different facets of this project and elements of data described, it may be helpful to narrow the focus so the key outcomes can be more prominent.

-The manuscript is very long. Please streamline where appropriate.

-occasional grammatical issues and typos throughout

Reviewer #3: Comments to the author:

General comment:

The topic is of interest and timely. However, major revisions are needed to make it publishable. Editing is needed to ensure clarity of writing.

Specific comments:

1.There are inconsistencies in between study objective, methods and presentation of results. Although the authors have mentioned that they have reported only the quantitative data, some of the findings are certainly not from quantitative data. For example, the results presented under ‘implementation challenges and solutions ‘and in Table 3.

On the other hand, at the end of ‘introduction’ section, the authors stated ‘’objective of the study was to implement a simplified antibiotic regimen (2 days gentamicin injection and 7 days oral amoxicillin) in a programme setting and to identify and document the challenges and opportunities for implementation at health post level in two regions in Ethiopia, Tigray and Jimma.’’ Organized, triangulated and sequential presentation of both the quantitative and qualitative findings may serve the purpose of this objective.

2.Under ‘ Context and setting of the study’ of the Methods section, the authors have mentioned “ the Primary Health Care Unit (PHCU) at the lowest level consists of five health posts, one health centre and one primary hospital. ……………” Later on, the authors have stated “A PHCU feeds into the secondary level of health care”. These are confusing. Need to be clear.

‘Specifically, they are expected to conduct postnatal home visits on days 1, 3, 7 and 42 as per the MNCH guideline.’ Who are they? Are they the HEWs or women from WDGs? Please make it clear.

3.Study design is not clear. Detailed description of the study design is needed for clear understanding.

4.First sentence under ‘study population’ of the Methods section needs to be rephrased.

5.First five sentences under ‘Intervention’ do not provide any information relating to intervention. These should go either under ‘introduction’ or study setting’. These are relating to Ethiopian health system and already briefly described under ‘introduction’ section, though.

6.Under ‘Ethical considerations’, the last sentence is not clear. It needs to be rephrased for clarity.

7.Data analysis section is not clear. It needs to be edited.

8.Before drawing a conclusion like ‘When referral is not feasible, PSBI management can be implemented safely and effectively at health posts by HEWs in the existing health system in Ethiopia’’, the author should perform analysis comparing the cases under different variables, between health posts and health centre, and show how statistically significant the results are. For example, compliance to treatment: health posts vs health centers; declared clinically treatment success: health posts vs health centers/hospitals

9.The authors should use a reference for making such a statement ‘Our data show that management of PSBI when referral is not feasible is possible in the Ethiopian context in line with data reported previously.’ (First sentence of the discussion section)

6. PLOS authors have the option to publish the peer review history of their article (what does this mean?). If published, this will include your full peer review and any attached files.

Reviewer #1: Yes: Tanya Guenther

Reviewer #2: No

Reviewer #3: No

---

## [Decision Letter · Decision Letter 1]

9 Oct 2020

PONE-D-20-08171R1

Innovative approach for potential scale-up to jump-start simplified management of sick young infants with possible serious bacterial infection when referral is not feasible: findings from implementation research

PLOS ONE

Dear Dr. Luel

Thank you for submitting your manuscript to PLOS ONE. After careful consideration, we feel that it has merit but does not fully meet PLOS ONE’s publication criteria as it currently stands. Therefore, we invite you to submit a revised version of the manuscript that addresses the points raised during the review process.

We look forward to receiving your revised manuscript.

Kind regards,

Lawrence Palinkas

Academic Editor

PLOS ONE

Additional Editor Comments (if provided):

Thank you for submitting a revised version of this manuscript. It is quite evident that you have devoted considerable effort to respond to the comments and suggestions provided by the reviewers of the original manuscript. Although the revised version is somewhat responsive to these comments, as you will note from their reviewers of the revised manuscripts, some significant issues remain. Of particular concern is the absence of baseline data and use of only descriptive statistics to reach your conclusions. Your rationale for doing so is not entirely satisfactory. The drawing of conclusions based on rates found in other countries remains subjective, especially since no comparisons were made between implementation experiences in Ethiopia and these other countries. We suggest you pay particular attention to these comments in considering whether to submit another revision. We also note that your justification of sample size also requires some explanation. Typically, a power analysis is provided to determine whether your sample is sufficiently large to test your hypotheses. Your response lacks any calculation of statistical power, perhaps because you conducted no statistical analysis.

Reviewers' comments:

Reviewer's Responses to Questions

**Comments to the Author**

1. If the authors have adequately addressed your comments raised in a previous round of review and you feel that this manuscript is now acceptable for publication, you may indicate that here to bypass the “Comments to the Author” section, enter your conflict of interest statement in the “Confidential to Editor” section, and submit your "Accept" recommendation.

Reviewer #2: (No Response)

Reviewer #3: (No Response)

2. Is the manuscript technically sound, and do the data support the conclusions?

Reviewer #2: Yes

Reviewer #3: No

3. Has the statistical analysis been performed appropriately and rigorously? 

Reviewer #2: No

Reviewer #3: No

4. Have the authors made all data underlying the findings in their manuscript fully available?

Reviewer #2: Yes

Reviewer #3: Yes

5. Is the manuscript presented in an intelligible fashion and written in standard English?

Reviewer #2: Yes

Reviewer #3: No

6. Review Comments to the Author

Reviewer #2: General –

- review grammar thoroughly, there are scattered missing articles (“the”, “an”) and incomplete sentences.

-Description of intervention planning and stakeholders is much improved.

-Manuscript is generally much easier to understand from an outside perspective.

-Were the interventions adapted/adjusted throughout the study period based on the data or was this system implemented at once and then data reviewed at the end of the study period? There are barriers and solutions described but it is not clear if these were determined based on data or subjective reporting of issues/concerns.

-You address barriers and solutions and that quality control was monitored monthly, however the data is only displayed as totals without showing change over time. How often was the data analyzed?

-Since there is no baseline data reported, you should explain why this is the case and that you used comparisons of other sites without your interventions. Even changes of the initial quarter to the final quarter could show improvement as the interventions were accepted and modified but there doesn’t seem to be any comparison over time. The treatment coverage rate referenced early in the paper is not restated compared to your results in the discussion to show that you achieved your aims. It is also not clear if this was a similar or the same community.

Since there is no direct baseline data, that is a significant limitation of the results that needs to be addressed in the discussion.

-Much better use of tables and figures to elaborate on information. Data is more completely described.

-The manuscript is still quite long – try to find ways to make wording more concise and eliminate any unnecessary detail.

Specific -

Line 75- An estimated 2.5 million neonatal deaths worldwide and 1 million deaths in Sub Saharan Africa 75occurred in 2018 contributing to 36% and 47% of under-5deathsrespectively

-Is the 1 million included in the 2.5 million or additional? Clarify.

Line 77 - Neonatal infections primarily bacterial in origin, including pneumonia, sepsis, and meningitis

-Sepsis is a clinical diagnosis that can result from a number of types of bacterial infections (including pneumonia and meningitis). Do you mean bacteremia? Clarify.

Line 110-113 – too much detail on farming/agriculture that is not applicable to the study

Line 133 – Previously stated that this is implementation research, however study design here says longitudinal study design.

Line 143 – incomplete sentence

Line 145-147 – There seems to be 2 primary outcomes – identification of 80% of infants and treatment of 80% of infants. Please make this statement clearer.

Secondary outcomes – did you have any targets for these?

Line 155 - swelling local to injection site

-Local swelling at injection site

Line 192- woreda is not a commonly known word

Line 228-231 – since this was not an issue, consider eliminating this section

Line 247 – CRF is defined later and should be defined with this first acronym use

Line 255-257 – redundant, information provided earlier and this is not specific to data collection

Line 260 – Earlier you discuss only Tigray, not Jimma.

Line 265 – I do not believe treatment failures is defined.

Line 261 – How often was data reviewed/analyzed? Was this done throughout the study period or retrospective on conclusion of the study period?

Line 288 – Was there a goal for the percent identified by HEWs? If so, please add target line to figure. I would write out sentence with your initial percent (8.4%) and final percent (29.2%) here in results in addition to demonstrating in figure.

Results – serious adverse effects was listed as a secondary outcome and this is not addressed in results

Figure 2 – The depiction of this data is confusing. It appears that this is the total number of cases of PSBI identified by the HEWs and how many were identified of this total through each quarter. It does not reflect the number appropriately identified or categorized which is the actual information of interest. I’m not sure you can interpret that a better identification rate occurred because there may have just been more cases during a particular quarter. Your denominator to say identification increased would have to include infants with delayed identification or identification by other means, showing how many the HEWs identified vs missed. You could instead show improvement in their skills based on the number per quarter that were misclassified based on review by the pediatrician, with this number hopefully decreasing as the intervention progressed.

Line 331 – You discuss the referral acceptance rate, however this was not included in your primary or secondary outcomes.

Line 360-361 – incomplete sentence

Line 428 – this is very subjective.

Implementation challenges and solutions should be part of the methods section. This is an essential part of your interventions and the description of the implementation process.

Reviewer #3: Thanks to the authors. However, still there are inconsistencies in between study objective, methods and presentation of results. The authors mentioned that the objective of this implementation research was to implement a simplified antibiotic regimen (2 days gentamicin injection and 7 days oral amoxicillin) for management of sick young infants with PSBI in a programme setting when referral was not feasible to achieve an overall treatment coverage of at least 80% in two districts in Tigray, Ethiopia (line 102-105). The objective of an implementation research is to produce evidence, not to implement a programme only.

Without having any counterfactuals and based only on descriptive statistical analysis, it is not sufficient to draw a conclusion that when referral is not feasible, young infants with PSBI can be managed appropriately at health posts by HEWs in the existing health system in Ethiopia. Rather the authors could explore the factors associated with the high coverage (identification and treatment of 87% young infants with PSBI) with inferential analysis, and with the support of more qualitative data.

The authors also mentioned that they collected both the quantitative and qualitative data and they also reported some qualitative findings. It was not mentioned in the method section how the qualitative data were collected and analysed.

7. PLOS authors have the option to publish the peer review history of their article (what does this mean?). If published, this will include your full peer review and any attached files.

Reviewer #2: No

Reviewer #3: No

---

## [Author Response · Author response to Decision Letter 1]

30 Nov 2020

Point by point responses to reviewers’ comments

Journal: PLOS ONE

Manuscript ID: PONE-D-08171

Manuscript Title: Innovative approach for potential scale-up to jump-start simplified management of sick young infants with possible serious bacterial infection when a referral is not feasible: findings from implementation research

Reviewer 2

1. General: 

Comment # 1

- review grammar thoroughly, there are scattered missing articles (“the”, “an”) and incomplete sentences.

Author’s response: 

Thank you. we have corrected the grammatical mistakes and the manuscript has been reviewed by a native English speaker.

Comment # 2

-Description of intervention planning and stakeholders is much improved. 

Author’s response: Thank you.

Comment # 3

-Manuscript is generally much easier to understand from an outside perspective.

Author’s response: Thank you.

Comment # 4

-Were the interventions adapted/adjusted throughout the study period based on the data or was this system implemented at once and then data reviewed at the end of the study period. There are barriers and solutions described but it is not clear if these were determined based on data or subjective reporting of issues/concerns.

Author’s response: 

Thank you. Since the study was implementation research, the interventions were adapted or adjusted based on the data collected. The sources of the data for the adjustment of interventions were quarterly review meetings, clinical mentoring, monitoring and evaluation checklist completed during the supportive supervision and follow up data collected using the CRFs (Case Reporting Forms). 

Comment # 5

-You address barriers and solutions and that quality control was monitored monthly, however, the data is only displayed as totals without showing change over time. How often was the data analyzed?

Author’s response: 

Thank you. We have presented our data in such a way that changes over time are displayed against the target and coverage (Please see Figure 1). The follow-up data from the case report forms were collected every month. The analysis was carried out quarterly. The field supervisor visited the health facilities every month for supportive supervision and clinical mentoring and provided on-site feedbacks to the problems encountered. 

Comment # 6

-Since there is no baseline data reported, you should explain why this is the case and that you used comparisons of other sites without your interventions. Even changes in the initial quarter to the final quarter could show improvement as the interventions were accepted and modified but there doesn’t seem to be any comparison over time. The treatment coverage rate referenced early in the paper is not restated compared to your results in the discussion to show that you achieved your aims. It is also not clear if this was a similar or the same community.

Author’s response: 

Thank you. The management of PSBI in young infants was not available in the study communities before the implementation of this research, therefore no comparative baseline data were available for comparison. But, the coverage of young infants with PSBI managed by the health extension workers (excluding those who directly went to a hospital) showed an improvement from 37% in the first quarter to 78% in the last quarter. We compared our findings with another study from Ethiopia referenced in our introduction [ref 15 in the list of references] . Also, our findings were in agreement with the AFRINEST study[ref 11 in the list of references] . The absence of direct baseline data is included as a limitation of this study as per the suggestion from the reviewer. 

Page 13 line 283 – 285 of the revised manuscript

Page 15, line 327 – 328 of the revised manuscript

Fig 1. Of the revised manuscript

Comment # 7

-Much better use of tables and figures to elaborate on the information. Data is more completely described.

Author’s response: Thank you. 

Comment # 8

-The manuscript is still quite long – try to find ways to make the wording more concise and eliminate any unnecessary detail.

Author’s response: 

Thank you. We have tried to make it as concise as possible.

2. Specific:

Comment # 1

Line 75- An estimated 2.5 million neonatal deaths worldwide and 1 million deaths in Sub Saharan Africa occurred in 2018 contributing to 36% and 47% of under-5 deaths, respectively

-Is the 1 million included in the 2.5 million or additional? Clarify.

Authors’ response

Thank you for the clarification. Yes, the one million deaths are included in the 2.5 million neonatal deaths and the sentence was re-written as follows.

In 2018, an estimated 2.5 million neonatal deaths occurred worldwide, of which 1 million were in Sub Saharan Africa contributing to 36% and 47% of under-5 deaths respectively.

Page 5; line 80 - 81 of the revised manuscript. 

Comment #2

Line 77 - Neonatal infections primarily bacterial in origin, including pneumonia, sepsis, and meningitis

-Sepsis is a clinical diagnosis that can result from a number of types of bacterial infections (including pneumonia and meningitis). Do you mean bacteremia? Clarify

Authors’ response

Thank you. Sepsis refers to the presence of bacteria in the blood with clinical manifestations. But, bacteremia is the presence of bacteria in blood. Hence it implies sepsis not bactermia.

Page 5; Line 82 -83 of the revised manuscript.

Comment #3

Line 110-113 – too much detail on farming/agriculture that is not applicable to the study.

Authors’ response

Thanks. We have removed the redundant information from the revised manuscript. The information was included to address the comments from another reviewer. 

Comment #4

Line 133 – Previously stated that this is implementation research, however study design here says longitudinal study design.

Authors’ response

Thank you. The research design is a longitudinal study design since data were collected prospectively from each young infant through a period of 20 months. But, the strategy used to implement the research design was implementation research. We have revised the text to make it clearer.

Page 7; line 135 of the revised manuscript. 

Comment #5

Line 143 – incomplete sentence

Authors’ response

Thank you for the comment. Addressed as per the reviewer’s comments. 

All babies born during the study period were included in the study. Young infants up to 2 months of age with one or more signs of PSBI (Panel 1) who were permanent residents of Raya Alamata and Raya Azebo districts were included. 

Page 7 ; line 147

Comment #6

Line 145-147 – There seems to be 2 primary outcomes – identification of 80% of infants and treatment of 80% of infants. Please make this statement clearer.

Secondary outcomes – did you have any targets for these?

Authors’ response

Thank you for the clarification. 

The sentence is rewritten as follows: The primary outcome of this study was identification of at least 80% of the PSBI cases and provision of appropriate treatment for at least 80% of those identified cases of PSBI in young infants aged up to 2 months. We did not have any targets for the secondary outcomes but we monitored their occurrences.

Page 7; line 149 - 151 of the Revised Manuscript.

Comment #7

Line 155 - swelling local to injection site

-Local swelling at injection site

Authors’ response

Thanks. The sentence was corrected as per the recommendation. 

Local swelling at the injection site

Page 8; Line 159- of the revised manuscript.

Comment #8

Line 192- woreda is not a commonly known word

Authors’ response

Thank you. We have replaced the word woreda by the district.

Pages10; line 196 of the revised manuscript. 

Comment #9

Line 228-231 – since this was not an issue, consider eliminating this section

Authors’ response

Thank you. Supplies and commodities section is removed from the manuscript.

Comment #10

Line 247 – CRF is defined later and should be defined with this first acronym use

Authors’ response

Thanks. Comment is well taken and corrected. Case reporting forms (CRFs)

Page 12; line 247

Comment #11

Line 255-257 – redundant, information provided earlier and this is not specific to data collection

Authors’ response

Thank you for the comment. Redundant information is removed.

Comment #12

Line 260 – Earlier you discuss only Tigray, not Jimma.

Authors’ response

Thank you. The sentence is removed. 

Comment #13

Line 265 – I do not believe treatment failure is defined.

Authors’ response

Thank you for this comment. Treatment failure was defined as clinical deterioration at any time and/ or failure to respond on day 4 of treatment.

Page 12; line 261 – 262 of the revised manuscript 

Comment #14

Line 261 – How often was data reviewed/analyzed? Was this done throughout the study period or retrospective on conclusion of the study period?

Authors’ response

Thank you. Data was reviewed frequently on arrival to the data management center at Mekelle University for its completeness and was analyzed quarterly for the review meetings. Complete data analysis was done at the end of the project.

Comment #15

Line 288 – Was there a goal for the percent identified by HEWs? If so, please add target line to figure. I would write out sentence with your initial percent (8.4%) and final percent (29.2%) here in results in addition to demonstrating in figure.

Authors’ response

Thank you.

Yes. The HEWs were expected to identify at least 80% of PSBI cases reported from the districts. We have modified figure 1 to include coverage from the target than the proportion. The coverage of identification of PSBI by the HEWs improved 37% in the first quarter to 78% in the last quarter. This coverage doesn’t include those who went directly to the hospital.

Page 13 line 280 – 281 of the revised manuscript

Page 15, line 323 – 324 of the revised manuscript

Figure 1 in the revised manuscript

.

 Comment #16

Results – serious adverse effects was listed as a secondary outcome and this is not addressed in results

Authors’ response

Thank you for the comment. It is included in the result section. No serious adverse effect or death was reported.

Page 14, line 288

Comment #17

Figure 2 – The depiction of this data is confusing. It appears that this is the total number of cases of PSBI identified by the HEWs and how many were identified of this total through each quarter. It does not reflect the number appropriately identified or categorized which is the actual information of interest. I’m not sure you can interpret that a better identification rate occurred because there may have just been more cases during a particular quarter. Your denominator to say identification increased would have to include infants with delayed identification or identification by other means, showing how many the HEWs identified vs missed. You could instead show improvement in their skills based on the number per quarter that were misclassified based on review by the pediatrician, with this number hopefully decreasing as the intervention progressed.

Authors’ response

Thank you. Figure 2 was included after a suggestion from reviewer 1 and has been accepted favourably. This figure intended to show the flow from livebirths to PSBI cases being identified and managed at different places of health care. We also documented their outcomes. We have revised figure 1 to provide information about the number of PSBI cases identified in each quarter. 

Comment #18

Line 331 – You discuss the referral acceptance rate, however, this was not included in your primary or secondary outcomes

Authors’ response

Thank you for the comment. It is included as a secondary outcome. 

Page 8; line 156 of the revised manuscript.

Comment #19

Line 360-361 – incomplete sentence

Authors’ response

Thank you for the comment. The incomplete sentence was removed.

Comment # 20

Line 428 – this is very subjective.

Authors response.

Thank you. This statement emanated from the findings presented in Figure 1. The coverage of PSBI identification during the early stages of the implementation was 30% and subsequently increased up to 62% due to the training provided to the HEWs and WDG leaders (Table 4). 

Comment # 21

Implementation challenges and solutions should be part of the methods section. This is an essential part of your interventions and the description of the implementation process.

Author’s response

Thank you for your comment. We, however, believe that the implementation challenges and solutions were a result of our implementation research and we couldn’t have anticipated these challenges and their solutions beforehand. 

Reviewer 3

Reviewer #3: Thanks to the authors. However, still there are inconsistencies in between study objective, methods and presentation of results. The authors mentioned that the objective of this implementation research was to implement a simplified antibiotic regimen (2 days gentamicin injection and 7 days oral amoxicillin) for management of sick young infants with PSBI in a programme setting when referral was not feasible to achieve an overall treatment coverage of at least 80% in two districts in Tigray, Ethiopia (line 102-105). The objective of an implementation research is to produce evidence, not to implement a program only.

Without having any counterfactuals and based only on descriptive statistical analysis, it is not sufficient to draw a conclusion that when referral is not feasible, young infants with PSBI can be managed appropriately at health posts by HEWs in the existing health system in Ethiopia. Rather the authors could explore the factors associated with the high coverage (identification and treatment of 87% young infants with PSBI) with inferential analysis, and with the support of more qualitative data.

The authors also mentioned that they collected both the quantitative and qualitative data and they also reported some qualitative findings. It was not mentioned in the method section how the qualitative data were collected and analysed.

Author’s responses

• Inconsistencies – In light of the comments, we have made the objective, methods and result sections consistent. The objectives are made to describe concisely what the research is trying to achieve (the results) through the approaches described in the method section. Implementation research in health is essential to improving the understanding of the challenges health systems face and how they impact implementation. Thus, in this implementation research, we have identified the contextual challenges affecting the implementation of PSBI in a program setting in two districts from Tigray, Ethiopia.

Thus the objective of this study was to implement a simplified antibiotic regimen (2 days gentamicin injection and 7 days oral amoxicillin) for management of sick young infants with PSBI in a programme setting when a referral was not feasible to identify at least 80% of PSBI cases, achieve an overall adequate treatment coverage of at least 80% and document the challenges and opportunities for implementation at the community level in two districts in Tigray, Ethiopia.

Page 6; line 105 – 110 of the revised manuscript 

• Data analysis – We used descriptive analysis for analysis. This intervention was new in this community and before the initiation of this implementation research, this service was not provided in these communities. Our data showed that high treatment coverage was achieved for young infants with signs of PSBI. If one compares with previously published data from elsewhere in Ethiopia of 50% coverage after the intervention [reference 15 cited as footnote 1 on page 2], our results demonstrate a higher treatment coverage of 87%. 

• Methods for qualitative data – Both quantitative and qualitative data were collected. But, in this manuscript, we primarily reported the quantitative findings. However, the results summarized in table 4 were qualitative data synthesized from the case report forms, review meetings, clinical mentoring visits and supportive supervision reports.

---

## [Editor Report · Decision Letter 2]

7 Dec 2020

Innovative approach for potential scale-up to jump-start simplified management of sick young infants with possible serious bacterial infection when referral is not feasible: findings from implementation research

PONE-D-20-08171R2

Dear Dr. Luel,

We’re pleased to inform you that your manuscript has been judged scientifically suitable for publication and will be formally accepted for publication once it meets all outstanding technical requirements.

Kind regards,

Lawrence Palinkas

Academic Editor

PLOS ONE
---

## [Editor Report · Acceptance letter]

25 Jan 2021

PONE-D-20-08171R2 

Innovative approach for potential scale-up to jump-start simplified management of sick young infants with possible serious bacterial infection when a referral is not feasible: findings from implementation research 

Dear Dr. Luel:

I'm pleased to inform you that your manuscript has been deemed suitable for publication in PLOS ONE. Congratulations! Your manuscript is now with our production department. 

Kind regards, 

on behalf of

Dr. Lawrence Palinkas 

Academic Editor

PLOS ONE